# ArtUV: Artist-style UV Unwrapping

**Yuguang Chen**
Sun Yat-sen University
Tencent Hunyuan
chenyg59@mail2.sysu.edu.cn

**Xinhai Liu**
Tencent Hunyuan
adlerxhliu@tencent.com

**Yang Li**
Tencent Hunyuan
liyang@mi.t.u-tokyo.ac.jp

**Victor Cheung**
Tsinghua University
Tencent Hunyuan
wikizhang@tencent.com

**Zhuo Chen**
Tencent Hunyuan
zhuooochen@tencent.com

**Dongyu Zhang** *
Sun Yat-sen University
zhangdy27@mail.sysu.edu.cn

**Chunchao Guo** *
Tencent Hunyuan
chunchaoguo@gmail.com

## Abstract

UV unwrapping is an essential task in computer graphics, enabling various visual editing operations in rendering pipelines. However, existing UV unwrapping methods struggle with time-consuming, fragmentation, lack of semanticity, and irregular UV islands, limiting their practical use. An artist-style UV map must not only satisfy fundamental criteria, such as overlap-free mapping and minimal distortion, but also uphold higher-level standards, including clean boundaries, efficient space utilization, and semantic coherence. We introduce ArtUV, a fully automated, end-to-end method for generating artist-style UV unwrapping. We simulates the professional UV mapping process by dividing it into two stages: surface seam prediction and artist-style UV parameterization. In the seam prediction stage, SeamGPT is used to generate semantically meaningful cutting seams. Then, in the parameterization stage, a rough UV obtained from an optimization-based method, along with the mesh, is fed into an Auto-Encoder, which refines it into an artist-style UV map. Our method ensures semantic consistency and preserves topological structure, making the UV map ready for 2D editing. We evaluate ArtUV across multiple benchmarks and show that it serves as a versatile solution, functioning seamlessly as either a plug-in for professional rendering tools or as a standalone system for rapid, high-quality UV generation.

## 1 Introduction

UV parameterization (Floater & Hormann, 2005; Sheffer et al., 2007), also known as UV unwrapping, is a fundamental task in computer graphics that plays an essential role in modern rendering pipelines. This process serves as the foundation for numerous downstream applications, including texture editing and lightmap generation. UV unwrapping specifically involves establishing a mapping from each 3D vertex $(x, y, z)$ of a mesh to corresponding 2D coordinates $(u, v)$ in UV space while maintaining the topological connectivity between vertices. To be effective in practice, a high-quality UV unwrapping requires both basic criteria such as low distortion and no overlaps, and higher-level criteria including clean boundaries, space efficiency, and semantic coherence.

---

*Corresponding author

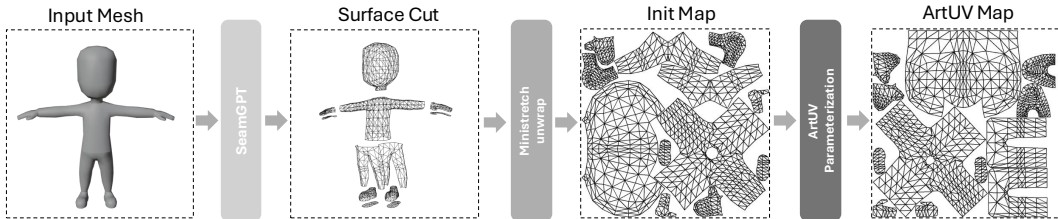

Figure 1: **ArtUV pipeline:** First, SeamGPT is used to predict the semantically meaningful seams on the input mesh surface. Based on these cutting line, the mesh is segmented into charts. For each chart, Ministretch-Unwrap (Rabinovich et al., 2017; Jiang et al., 2017; Lévy et al., 2023) is used to quickly generate the initial UV map. This initial UV map is then input into the ArtUV parameterization module to Optimize the artist-style UV map.

UV unwrapping methods can typically be categorized into three types: top-down, bottom-up (Sorkine et al., 2002; Li et al., 2018), and learning-based (Srinivasan et al., 2024; Zhang et al., 2024). The top-down method begins by identifying seams on the 3D object to partition the surface into separate charts. These individual charts are then unwrapped with careful attention to minimizing distortion while preserving neat boundaries. The process concludes by efficiently packing these UV islands into a cohesive, complete UV map. Conversely, the bottom-up approach starts with the entire object surface as discrete elements, then progressively merges these triangles by optimizing an energy function until convergence produces the final UV map. Learning-based methods employ unsupervised training through a cyclic mapping network. This architecture projects the 3D object into UV space and reconstructs it back to 3D coordinates, forming a complete round-trip transformation. During this process, physical constraints such as mapping bijectivity guide the network optimization. Once trained, the forward 3D-to-2D pathway directly generates the UV unwrapping.

Traditional top-down methods demand considerable time and expertise, requiring experienced artists to determine optimal cutting locations and manually adjust UV islands after unwrapping. This reliance on manual intervention makes it challenging to achieve the optimal balance between distortion and neatness. Bottom-up methods face a different issue: their discretization and re-clustering processes often produce fragmented UV maps that compromise the neatness of individual UV islands. Meanwhile, current learning-based methods require extensive training time for each 3D model and typically rely on point cloud inputs to handle diverse formats. This approach destroys the topological relationships between points, creating chaotic UV mappings with significant overlap that severely limits practical usability. More fundamentally, both bottom-up and learning-based methods lack semantic awareness. For character models, artists prefer semantically meaningful divisions where limbs and torso are unwrapped separately, as this greatly facilitates subsequent texture editing tasks. Without this semantic understanding, these automated methods fail to produce the intuitive, artist-friendly results that practical workflows demand.

In this paper, we introduce a learning-based top-down method for UV unwrapping, decomposing the task into surface seam prediction and UV parameterization. For seam prediction, we leverage SeamGPT (Li et al., 2025) to semantically segment the original mesh surface. Then, in the UV parameterization component, we employ an Auto-Encoder model to simulate the manual adjustment process of UV mapping. Specifically, we encode the original mesh information and constrain the output UV map in terms of its neatness, overlap, and distortion. After extensive training with carefully curated data, we obtain a general UV unwrapping method capable of optimizing the initial UV map from traditional modeling software into one that aligns with artist-style UV mapping. Ultimately, we present a fully automated, end-to-end UV unwrapping method that generates high-quality, artist-style UV maps with semantic information in seconds.

Extensive experiments demonstrate that our method outperforms traditional and learning-based UV unwrapping methods. Our main contributions are as follows:

- We propose ArtUV, a novel learning-based framework that automatically generates artist-style UV maps from 3D meshes with semantic awareness and professional quality standards.
- We present ArtUV-200K, a high-quality dataset comprising 200K artist-style UV maps, filling the gap in datasets specifically focused on high-quality UV maps.

- Comprehensive evaluations demonstrate that ArtUV significantly outperforms state-of-the-art UV unwrapping methods across distortion, utilization, and processing speed metrics.

## 2 RELATED WORK

### 2.1 OPTIMIZATION BASED UV UNWRAPPING METHOD

Traditional UV unwrapping methods primarily rely on numerical optimization techniques that aim to minimize specific energy functions when mapping 3D mesh surfaces to 2D parameter domains, forming the core of both early and many contemporary unwrapping tools. These methods can be classified into two categories based on whether surface seams are pre-defined: For methods with given seams that partition the mesh into separate charts, LSCM (Lévy et al., 2023) and ABF++ (Sheffer et al., 2005) optimize angles in the 2D domain to approximate their 3D counterparts, theoretically achieving conformal mapping with minimal angular distortion, though often struggling to guarantee global bijectivity. Conversely, approaches such as SLIM (Rabinovich et al., 2017), SCAF (Jiang et al., 2017), and Smith & Schaefer (2015) initialize UV maps as bijective mappings and explicitly incorporate local/global overlap constraints during optimization to maintain bijectivity. While these methods typically produce low-distortion results, they face inherent challenges in simultaneously optimizing for both boundary regularity and distortion minimization. For methods without pre-defined seams that jointly optimize surface cutting and parameterization, (Sorkine et al., 2002) discretizes surfaces into compact triangular patches that iteratively grow around seed triangles until meeting termination criteria, while OptCuts (Li et al., 2018) proposes a joint energy function minimizing both seam length and distortion for synchronous optimization. However, such joint optimization algorithms often yield impractical UV maps due to excessive fragmentation or lack of semantic coherence. Consequently, current production pipelines still require substantial manual intervention, including hand-placed seams and manual UV island adjustments, to meet professional rendering requirements.

### 2.2 LEARNING BASED UV UNWRAPPING METHOD

With the advancement of deep learning techniques, learning-based UV unwrapping methods have emerged as a research focus. Nuvo (Srinivasan et al., 2024) employs a multi-category neural network architecture to separately handle mesh segmentation and parameterization, utilizing multiple loss functions to enforce bijectivity and minimize distortion. FAM (Zhang et al., 2024) introduces a physically-inspired framework comprising sub-networks for surface cutting, UV deformation, unwrapping, and packing, achieving point-to-point mapping for arbitrary 3D representations through a bidirectional cyclic mapping mechanism. However, these methods exhibit notable limitations: the absence of semantic and requiring time-consuming per-scene optimization. Particularly, FAM's point-cloud-based approach disrupts topological structures and generates irreparable overlapping artifacts. Most critically, these algorithms struggle to achieve the neatness and aesthetic quality of artist-style unwrapping, resulting in suboptimal UV utilization. As a result, they are impractical for use in professional rendering pipelines.

### 2.3 CUTTING-SEAM PREDICTION METHOD

The UV unwrapping process typically involves surface cutting and parameter mapping. Recent advances in deep learning have inspired several works (Wang et al., 2020; Bazazian & Parés, 2021; Himeur et al., 2021) that employ neural networks for edge point detection, framing it as a per-point classification task. EC-Net (Yu et al., 2018) reformulates this approach as a regression problem by learning residual point coordinates and point-to-edge distances to identify edge points more precisely. Building upon these developments, SeamGPT (Li et al., 2025) innovatively simulates professional workflows by utilizing an autoregressive network to model surface cutting as a next-cut-point prediction task, enabling semantically meaningful cuts for both artist-created and AI-generated meshes.

## 3 DATA PREPARATION

We collected UV-mapped mesh models from multiple open-source 3D datasets, including Objaverse (Deitke et al., 2022), Objaverse-XL (Deitke et al., 2023), and 3D-FUTURE (Fu et al., 2021), focusing on artist-style UV maps. To ensure high-quality training data, we implemented a rigorous filtering process. From an initial collection of approximately 350,000 textured mesh models, we first decomposed the complete models into individual UV islands and then filtered out cases with overlapping UV maps or excessive fragmentation (fewer than 5 vertices per island), yielding a refined subset of around 300,000 independent UV islands. As illustrated in Figure 2, vertex count distribution analysis indicated that over 97% of these islands contained fewer than 500 vertices; larger islands were excluded

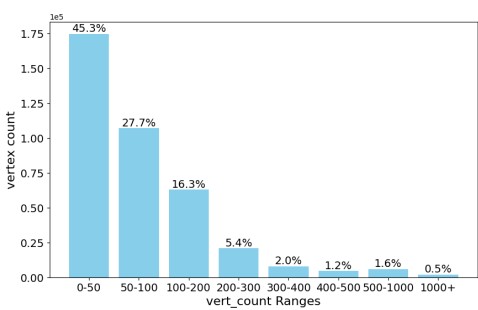

Figure 2: Distribution of vertex count in ArtUV-200K.

to enhance training stability and mitigate memory overhead. We then manually selected UV islands with meaningful semantic information and well-organized layouts. Each candidate was processed using Ministretch-Unwrap method and evaluated against the original UV maps using SSIM (Sara et al., 2019), with islands scoring between 0.5 and 0.8 selected as high-quality, manually adjusted cases.

Ultimately, we construct ArtUV-200K, a high-quality benchmark dataset for artist-style UV unwrapping, containing approximately 15,000 objects and 200k UV islands.

## 4 METHOD

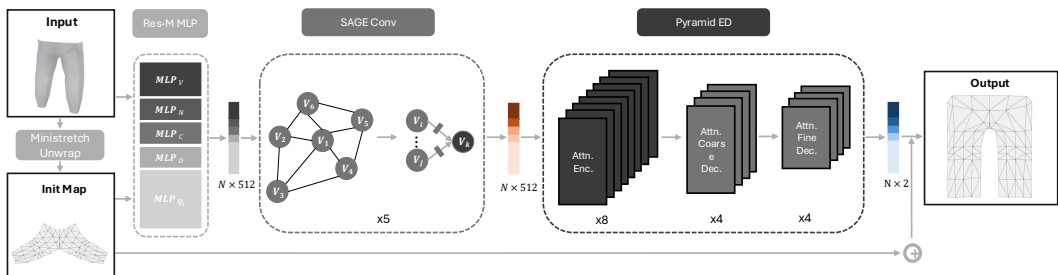

Figure 3: **ArtUV parameterization architecture**. Res-M MLP: Performs importance-based dimension mapping of input parameters; SAGEConv: Fuses local features between adjacent vertices via graph convolution; Pyramid ED: Enables global vertex interaction through attention encoder and coarse-to-fine decoder; Output: Combines predicted offsets with initial map for final UV parameterization.

We present ArtUV, a novel automatic artist-style UV unwrapping approach that incorporates both surface cutting and UV parameterization. For the surface cutting, we replicate SeamGPT, an advanced auto-regressive (Lütkepohl, 2013) seam prediction model. For UV parameterization, we propose a groundbreaking artist-style UV parameterization model based on an Auto-Encoder.

In the following sections, Sec. 4.1 will first provide a concise overview of SeamGPT. Next, in Sec. 4.2, we present our artist-style UV parameterization method, where we formulate the artist-style UV unwrapping task as learning the discrepancy between traditional software-generated UV maps and artist-optimized UV maps. Finally, in Sec. 4.3, we will focus on the loss function in parameterization module, which enables the model to generate UV maps that are as neat and low-distortion as those created by artists.

## 4.1 PRELIMINARIES

Given a 3D mesh model $M$, which includes a vertex set $V \in R^{N \times 3}$ and a triangle face set $F \in R^{M \times 3}$, the surface cutting task is identify cut vertices $V_c \subseteq V$ that form connected seams along mesh edges, partitioning the surface into discrete charts. SeamGPT formulates surface cutting as a sequence prediction problem by spatially sorting and quantizing cut vertices, where each token represents a coordinate value and six consecutive tokens define a seam segment. Specifically, for an input 3D mesh, SeamGPT first samples point clouds on vertices and edges and compresses them into a latent shape condition using a point cloud encoder from Zhao et al. (2025). Next, following Hao et al. (2024), SeamGPT builds an hourglass-like autoregressive decoder with multiple Transformer (Vaswani et al., 2017) stacks at each level, which bridges these hierarchical stages through causality-preserving downsampling and upsampling layers. The decoder autoregressively generates coordinate tokens starting from SOS until EOS, with final seam vertices obtained by nearest-point projection of discrete tokens onto the mesh surface.

## 4.2 ArtUV PARAMETERIZATION

**Modeling.** UV parameterization aims to establish a continuous bijective mapping from a 3D mesh $M$ to a 2D plane $P \in [0,1]$. The corresponding point set on the plane is represented as $Q \in R^{N \times 2}$. The most straightforward approach would be to directly use all mesh information ($I_M = \{V, F, N, C, D\}$), including vertices $V$, faces $F$, normals $N$, degrees $C$ and curvature $D$, as input and we could then train a powerful model to learn the 3D-to-2D mapping. However, extensive empirical evidence has shown that despite having groundtruth, directly learning the mapping process is a challenging task. Moreover, a simple projection is not our goal; our ultimate aim is to obtain a neat and low distorted artist-style UV map. Therefore, inspired by traditional top-down modeling process, we utilize the UV map $Q_i$ obtained through an optimization-based unwrapping method as initialization, enabling the parameterization model to learn how to adjust $Q_i$ into an artist-style UV map. We additionally include $Q_i$ as part of the input and the final input represented as $I = I_M \cup Q_i$. Next, we use the model to predict the offset $Q_o$ required for each vertex in the initial UV map during the manual adjustment process. By adding the predicted offset $Q_o$ to $Q_i$, we obtain the final UV map $Q_{pred} = Q_i + Q_o$, which satisfies the artist's standards.

**Architecture.** The architecture of our parameterization model is illustrated in Figure 3. First, we devise a residual MLP module (Res-M MLP) with adaptive dimension mapping, which dynamically adjusts feature dimensions according to their empirically observed importance in UV unwarpping tasks ($Q_i > V > C = N = D$), where the residual structure effectively preserve essential input information while enhancing feature representation. To maintain topological consistency, we construct a graph structure with vertices as nodes and face adjacency as edges, employing SAGE-Conv (Hamilton et al., 2017) for local feature propagation among neighboring nodes. The processed features are then fed into a Pyramid ED module, where stacked attention layers in the encoder enable global vertex feature interaction, followed by a coarse-to-fine decoder that concurrently extracts coarse-grained global structure and fine-grained local details to predict UV space offsets $Q_o$. This predicted offset is then added to the initial UV coordinates $Q_i$, resulting in the final predicted UV map $Q_{pred}$.

## 4.3 LOSSES

To generate UV maps that align with the artist's design standards—namely neatness, minimal distortion, and free overlap, we optimize the UV parameterization model's performance through a multi-term weighted loss function.

The fundamental component is the UV reconstruction loss, which directly measures the positional discrepancy between predicted UV coordinates $Q_{pred}$ and groundtruth coordinates $Q_{gt}$. Before computing this loss, a critical preprocessing step is performed to align the initial UV coordinates $Q_i$ with the groundtruth $Q_{gt}$ in rotation space. As in Figure 4, this alignment is achieved using Horn's method (Horn, 1987), which computes an

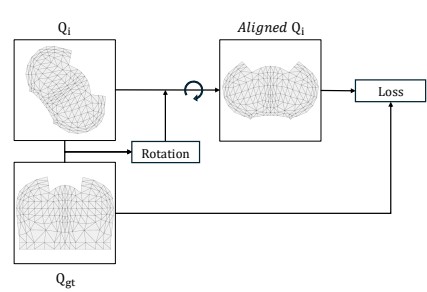

Figure 4: UV map alignment.

optimal rotation matrix $R$ to align the two sets of coordinates by minimizing their rotational discrepancy. Specifically, The covariance matrix $W$ between $Q_i$ and $Q_{gt}$ is computed as:

$$W = \sum_{i=1}^{N}(q_i - \bar{q}) \cdot (p_i - \bar{p}) \tag{1}$$

where $q$ and $p$ are the corresponding points in $Q_i$ and $Q_{gt}$ and $\bar{q}$ and $\bar{p}$ are their mean values.

Then, Singular Value Decomposition(SVD) is applied to $W$ to obtain the optimal rotation matrix $R$:

$$W = U\Sigma V^T \tag{2}$$

$$S = \begin{bmatrix} 1 & 0 \\ 0 & sign(\det(U) \cdot \det(V^T)) \end{bmatrix} \tag{3}$$

$$R = USV^T \tag{4}$$

This rotation matrix $R$ is used to align the initial coordinates $Q_i$ with the ground truth coordinates $Q_{gt}$. After the alignment, the reconstruction loss is computed as the pointwise $L_1$ distance between the predicted and ground truth coordinates:

$$L_{recon} = \|Q_{gt} - Q_{pred}\|_1 \tag{5}$$

To further enhance the structural regularity of predicted UV maps to match manually unwrapped results, we introduce an edge-preserving silhouette loss. This is implemented through differentiable rendering of UV map silhouette as in Figure 5. By computing the $L_2$ distance between silhouette maps of predicted and ground truth UV map, the model is guided to focus on boundary information that significantly reflects the neatness of UV islands, expressed as:

$$L_{silhouette} = \|Render_{gt} - Render_{pred}\|_2 \tag{6}$$

Subsequently, we optimize the distortion of the predicted UV map by adding a distortion loss term. Specifically, for each triangular face $f$, we compute the Jacobian matrix of its 3D-to-2D mapping, and after performing Singular Value Decomposition, we obtain the singular values $\sigma^1$ and $\sigma^2$, which characterize the stretching intensity. The distortion loss function is defined as the mean of the absolute differences of the singular values across all faces, which approaches zero in the case of an ideal conformal mapping.

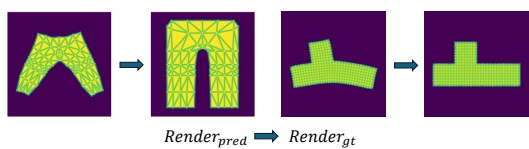

$$Render_{pred} \Longrightarrow Render_{gt}$$

Figure 5: UV map silhouette.

$$L_{distortion} = \frac{1}{\sum_{f \in \mathcal{F}}|A_f|} \sum_{f \in \mathcal{F}} |A_f| \left\|\sigma_f^1 - \sigma_f^2\right\|_1, \tag{7}$$

where $\mathcal{F}$ is the set of all triangles on the input surface and $|A_f|$ is the area of $f$.

Finally, to prevent UV map overlaps that would adversely affect subsequent texture mapping processes, we implement an overlap penalty based on normal direction. The key observation is that overlapping triangular faces in the UV domain exhibit flipped normal directions. We formulate this constraint by introducing a penalty term proportional to the count of faces with negative normal directions:

$$L_{overlap} = \sum_{f \in F}(n_f \cdot \vec{z} < 0) \tag{8}$$

where $F$ represents faces in the UV map, $n_f$ is the normal vector of face $f$ in the UV map, and $\vec{z}$ is the reference viewing direction.

Consequently, The complete objective function combines these loss terms through a weighted linear combination:

$$L_{\text{total}} = \omega_r L_{\text{recon}} + \omega_s L_{\text{silhouette}} + \omega_d L_{\text{distortion}} + \omega_o L_{\text{overlap}} \tag{9}$$

where $\omega_i, (i = r, s, d, o)$ represents the weight for each corresponding loss component.

## 5 EXPERIMENT

We conduct both qualitative and quantitative evaluations of our model on the ArtUV-200K and the FAM benchmark. First, we compare the UV unwrapping performance of our parameterization model with that of mainstream professional modeling software. Next, we compare our complete ArtUV method with some advanced algorithms, covering the entire process from surface segmentation to UV parameterization. Finally, we perform ablation studies to assess the rationale and effectiveness of our loss function design from multiple perspectives.

### 5.1 IMPLEMENTATION DETAILS

**SeamGPT.** We consulted the authors of SeamGPT for detailed dataset and model implementation specifics, and successfully reproduced the complete SeamGPT model. For additional details, please refer to Supplementary Materials.

**ArtUV parameterization.** We use Blender's ministretch algorithm as the initialization method for the UV parameterization module. The Res-M MLP module map the feature dimensions of the input UV, vertices, normals, curve, and degree to 128, 62, 32, 32, and 32. These features then passed through 5 SageConv layers, resulting in a 512 dimensional graph feature. The extracted features are processed by an attention encoder with 512 dimensions, 8 heads, and 8 layers. The coarse-to-fine decoder then down-samples the features by factors of 1/2 and 1/4. Finally, the output mapping layer predicts the UV coordinates (dimensionality 2) using a Tanh activation function, ensuring the predictions remain within the range $[-1, 1]$. The model is trained on 24 H20 GPUs (96 GB) for 700K steps with a batch size of 32. In the loss function, the weights $\omega_r$, $\omega_s$, $\omega_d$ and $\omega_o$ are set as 1.0, 1.0, 0.0001 and 0.01. Inference can be performed on most consumer-grade GPUs, with memory usage not exceeding 10 GB when the model contains fewer than 1000 faces.

### 5.2 COMPARED WITH PROFESSIONAL SOFTWARE

We first compare our UV parameterization model with current professional modeling software in the ArtUV-200K, which contains 100 diverse 3D models, each with artist-manually marked seams. This allows us to eliminate any interference from seams quality and directly compare the unwrapping results. We conduct quantitative evaluations in terms of mesh distortion and UV utilization. Specifically, the mesh distortion is computed as the average conformal energy over all triangular faces of the mesh. For UV utilization, we apply the UVPackMaster plugin for layout optimization in all unwrapping algorithms.

| Method | Distortion↓ | Utilization(%)↑ | Artist-Level↑ |
|---|---|---|---|
| Blender | 9.85 | 62.74 | 3.34 |
| Maya | 9.66 | 67.53 | 1.32 |
| 3DsMax | 11.88 | 67.01 | 1.53 |
| Artist-manual | 10.90 | 70.08 | 4.12 |
| **Ours** | **9.52** | **72.57** | **4.22** |

Table 1: Quantitative results on ArtUV-200K benchmark.

The metrics are presented in Table 1, where our algorithm outperforms in all evaluated criteria. Compared to professional modeling software, our algorithm significantly improves UV utilization while maintaining low distortion. Even when compared to artist-manually unwrapping results, our approach exhibits better performance in both distortion and utilization, indicating that our algorithm achieves a level of perfect balance between distortion and neatness that is difficult for conventional algorithms or manual to attain. Furthermore, using the same texture reference, we apply the texture generation method (Zhao et al., 2025) to UV maps produced by different approaches. The visualizations in Figure 6 show that our unwrapping method produces more Horizontal and vertical UV

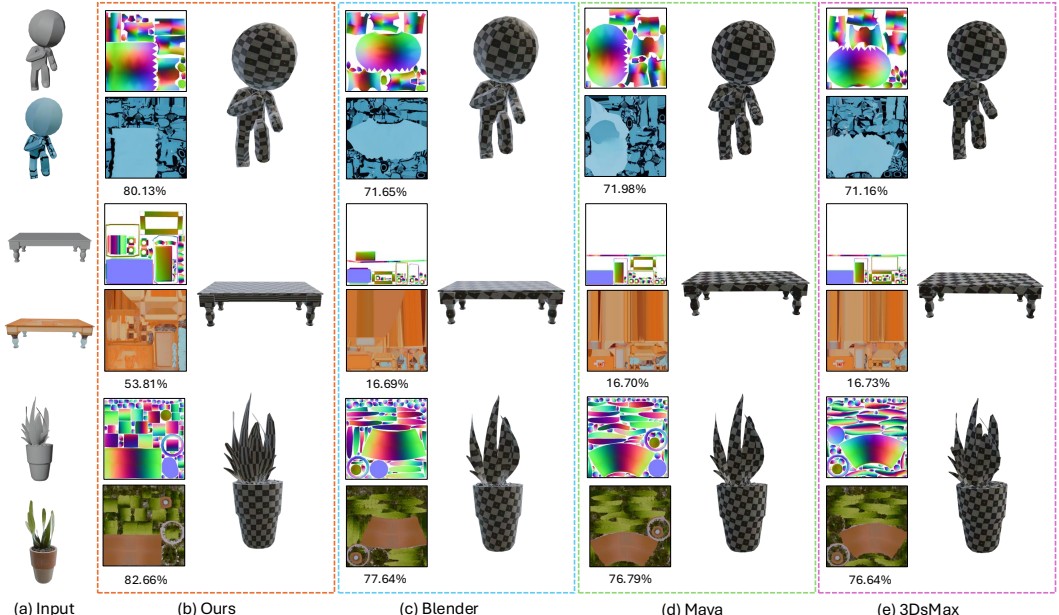

Figure 6: Qualitative Result on ArtUV-200K. (a) The input mesh and reference texture image. (b)-(e) Results of our method, along with Blender, Maya, and 3DsMax. For each method, the upper-left image shows the UV unwrapping visualized using normal directions, the lower-left image displays the texture map based on the reference texture, with the value below the texture map representing the UV utilization rate, and the right image presents the 3D model with a checkerboard texture.

maps. This not only leads to a substantial increase in UV utilization, but also makes the texture map clearer, providing significant convenience for subsequent tasks such as texture editing.

Moreover, because the quality of artistic style is inherently subjective, we conducted a user study to further evaluate the Artist-Level of our results. We randomly selected 10 representative cases from the ArtUV-200K and invited 30 professional 3D artists to score the artistic style of the generated UV maps on a five-point Likert scale, where 5 indicates results most similar to artist-created UV maps and 0 represents no resemblance to artistic style. As shown in the last column of Table 1, our method even slightly surpass those produced by manual artist adjustments, demonstrating its strong capability to capture the intended artistic style in UV unwrapping.

## 5.3 COMPARED WITH SOT ALGORITHM

We then evaluate our fully automated ArtUV method using the FAM benchmark (which lacks seam information), comparing it with three algorithms: XAtlas, Nuvo, and FAM. The comparison covers the entire UV unwrapping process, including surface segmentation and UV parameterization. Quantitative evaluations in Table 2 based on mesh distortion, computational runtime and UV islands count demonstrated that our algorithm outperformed the others. It is evident that XAtlas

| Method | Distortion↓ | Runtime(s)↓ | Fragments↓ |
|--------|-------------|-------------|------------|
| XAtlas | 9.44 | 80.4 | 1292 |
| Nuvo | 32.24 | 2925.8 | 1 |
| FAM | 76.28 | 5656.3 | 1 |
| **Ours** | **8.91** | **36** | 14 |

Table 2: Quantitative results on FAM benchmark.

produces overly fragmented and semantically poor segments, while Nuvo and FAM suffered from prolonged computation times due to requiring per-model training. Furthermore, as shown in Figure 7, the UV unwrapping results from Nuvo and FAM, characterized by disorganized topology, are impractical for professional rendering pipelines, making a comparison of UV utilization unnecessary as well. In contrast, our ArtUV method demonstrates outstanding performance, providing

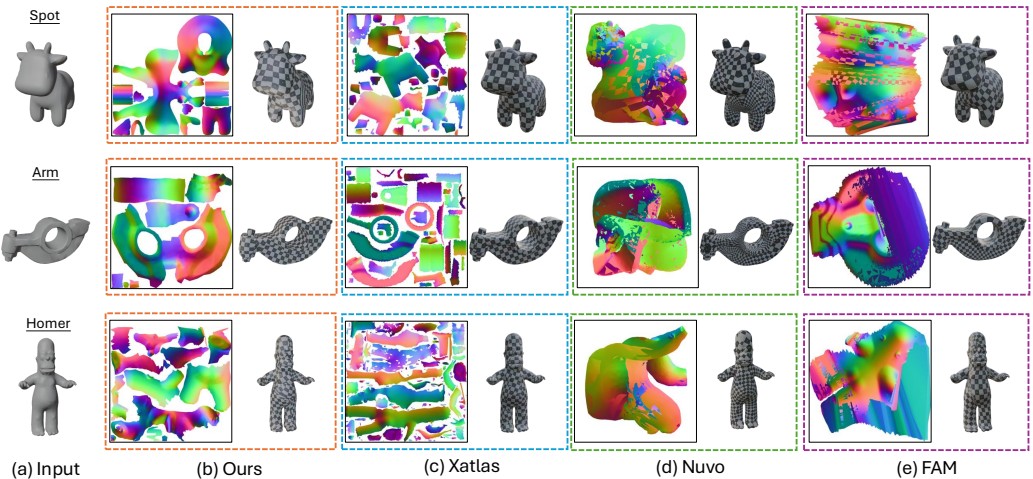

Figure 7: Qualitative Result on FAM benchmark(Spot, Arm, and Homer). (a) shows the input mesh from the FAM benchmark, while (b) to (e) present the UV unwrapping results for our method, XAtlas, Nuvo, and FAM, respectively. It is clear that our algorithm directly produces elegant and high-quality UV maps. In contrast, XAtlas results in overly fragmented and semantically weak UV maps, while Nuvo and FAM generate disordered and unusable UV maps.

an end-to-end solution that directly generates UV maps with low distortion, organized layout, and semantic information, fully meeting the design standards of artists.

## 5.4 ABLATION STUDIES

We conducted several ablation studies to evaluate the contributions of our designed loss functions.

**Silhouette Loss.** As shown in Figure 8 (a) and Table 3, when the silhouette loss is omitted, the model fails to optimize the boundaries, resulting in less aligned and structured UV maps, which in turn leads to lower UV utilization and reduced Artist-Level scores.

**Overlap Loss.** Including the overlap loss suppresses the model's tendency to generate overlapping UV faces. As shown in Figure 8 (a), the UV maps are more orderly with this loss. Table 3 further shows a significant reduction in overlapping face when the overlap loss is applied.

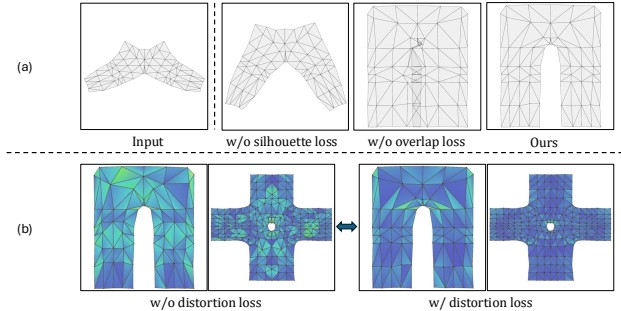

Figure 8: Qualitative ablation results.

**Distortion Loss.** Figure 8 (b) visualizes UV distortion using a color-coded scheme, where brighter yellow regions indicate higher distortion. Quantitative results in Table 3 demonstrate that adding the distortion loss adjusts internal UV coordinates to more reasonable positions, effectively reducing distortion.

## 6 CONCLUSION

In this paper, we propose ArtUV, an end-to-end method for generating artist-style UV maps. We decompose the problem into surface segmentation using SeamGPT and UV parameterization that predicts offsets from initial UV maps to artist-style results. Our method generates neat, well-organized

| Loss | Distortion ↓ | Loss | Overlap (%) ↓ | Loss | Utilization (%) ↑ | Artist-Level ↑ |
|------|------------|------|-------------|------|-----------------|--------------|
| w/o Dist. | 10.56 | w/o Ovlp. | 29.0 | w/o Sil. | 64.33 | 3.67 |
| w/ Dist. | **9.52** | w/ Ovlp. | **0.0** | w/ Sil. | **72.57** | **4.12** |

Table 3: Quantitative ablation results.

UV maps with low distortion in seconds, addressing current issues of long processing times and lack of semanticity in professional workflows. Extensive experiments demonstrate that ArtUV outperforms existing approaches across multiple metrics, holding significant potential for efficiency improvements in downstream applications.

**Limitation.** Our current approach exhibits two key limitations. First, the method's performance is highly sensitive to the quality of surface cutting. Incomplete or inaccurate seams may cause severe distortions during UV initialization, resulting in significant internal deformation despite the output maintaining clean edges. Second, our pipeline does not yet support UV island reuse, since imperfect alignment of reused islands might cause serious overlapping artifacts. Additionally, island reuse may introduce additional complexity to model training. Future work will focus on: (1) enhancing the quality and stability of seams (e.g., applying secondary segmentation to high-distortion areas), and (2) integrating UV island reuse into the pipeline (e.g., via similarity-based merging of optimized islands).

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

## A APPENDIX

### A.1 MORE RESULTS

#### A.1.1 QUANTITATIVE RESULTS ON FAM BENCHMARK.

As shown in Table 4, we provide a detailed comparison of our method with XAtlas, Nuvo, and FAM in terms of the distortion metrics for each category in the FAM-benchmark.

| Method | XAtlas | Nuvo | FAM | **Ours** |
|---|---|---|---|---|
| Bimba | 15.44 | 19.12 | **12.10** | 20.02 |
| Lucy | **0.011** | 57.894 | 35.13 | 0.043 |
| Ogre | **0.66** | 26.22 | 11.55 | 0.75 |
| Armadillo | **0.17** | 114.21 | 59.87 | 0.3492 |
| Bunny | 61.83 | 16.84 | **7.33** | 58.19 |
| Nefertiti | **0.026** | 20.92 | 11.2 | 0.23 |
| Dragon | 0.22 | 61.02 | 904.89 | **0.12** |
| Planck | 0.14 | 11.09 | 4.67 | **0.062** |
| Homer | **7.51** | 21.92 | 14.19 | 19.20 |
| Teapot | **2.42** | 17.56 | 8.77 | 3.06 |
| Cheburashka | **8.41** | 19.75 | 12.21 | 10.86 |
| Spot | 12.77 | 12.93 | 9.37 | **8.73** |
| Arm | 29.98 | 37.34 | 20.98 | **8.54** |
| Beast | **0.062** | 34.19 | 23.54 | 1.38 |
| Cow | 1.94 | 12.70 | 8.49 | **1.52** |
| Avg. | 9.44 | 32.24 | 76.28 | **8.91** |

Table 4: Quantitative results on FAM benchmark using the face distortion metric.

### A.1.2 Qualitative Result on ArtUV-200K benchmark.

As shown in the Figure 9, we present more UV unwrapping results from the ArtUV-200K benchmark. It is evident that the UV maps obtained using our method are more elegant and well-organized.

### A.1.3 Results of different initialization methods.

Our ArtUV parameterization module can be seamlessly integrated as a plugin with various professional modeling software. To demonstrate its robustness, we replaced Blender's initial UV unwrapping results with Maya and 3DsMax based initialization. As shown in Figure 10, our method consistently produces high-quality, artist-style UV maps regardless of the initialization approach used.

### A.2 More Implemention Details of SeamGPT

We consulted the authors of SeamGPT for detailed dataset and model implementation specifics, and successfully reproduced the complete SeamGPT model. Begin with a targeted point sampling strategy that collects a total of 61,440 points—evenly split between 30,720 points on vertices and 30,720 points along edges. Then, we implement a hierarchical hourglass-style decoder defined by a three-level abstraction structure with depth configuration (2, (4, 12, 4), 2), where each number represents the number of transformer blocks at that level. Each block has 1,536 dimensions and 16 attention heads, incorporating 10-bit quantized positional encoding for sequences up to 36,864 tokens. The model is trained on 64 H20 GPUs(96GB) for 200k steps with a fixed learning rate of 1e-4, gradient clipping at 0.5 and a batch size of 128. Data augmentation techniques including random scaling with $[0.95, 1.05]$, random vertex jitter with noise level 0.01, and random rotation are implemented to improve model robustness during the training process.

### A.3 Code

We provide our demo inference and model code as part of our supplementary materials. All source code and pretrained models will be made publicly available upon acceptance.

### A.4 Dataset

ArtUV-200K benchmark is tentatively planned for public release upon paper acceptance. At the same time, we will release our data processing scripts including UV island segmentation and data filtering to assist researchers in curating high-quality UV data from proprietary datasets.

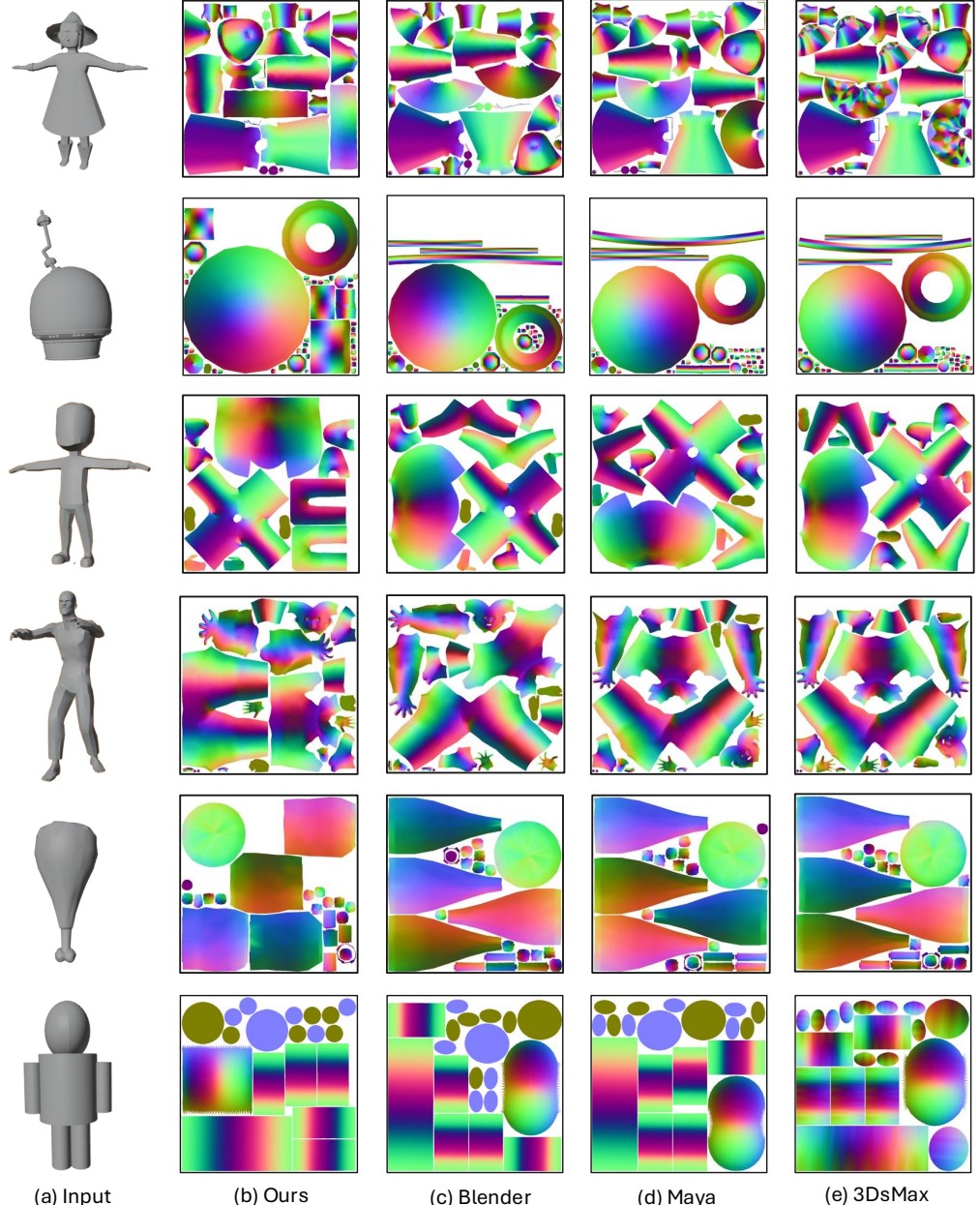

Figure 9: More Qualitative Results on ArtUV-200K.

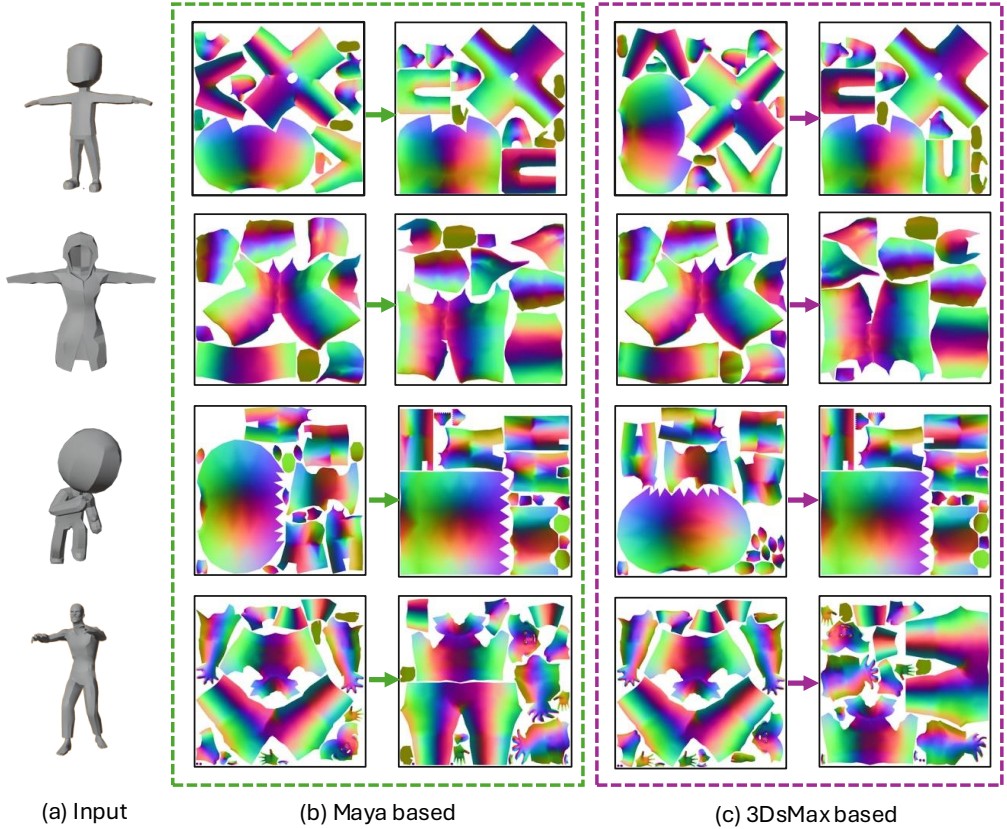

(a) Input          (b) Maya based          (c) 3DsMax based

Figure 10: Results of different initialization methods.(a) Input original mesh; (b) and (c) display unwrapping results based on Maya initialization and 3DsMax initialization, respectively. For each method, the left sub-figure shows the initial unwrapping result, while the right sub-figure presents the optimized output after processing through our ArtUV parameterization module.

