# OpenReview forum: "ArtUV: Artist-style UV Unwrapping"
_ICLR.cc/2026/Conference — ICLR 2026 Poster_

### Official Review · Reviewer_ZizJ · 2025-10-29

**Soundness:** 4
**Presentation:** 3
**Contribution:** 3
**Rating:** 8
**Confidence:** 5

**Summary:**

The authors address the challenge of artist-level UV unwrapping. To achieve this, they begin by reimplementing SeamGPT to partition the mesh into distinct, semantically meaningful islands. An initial UV unwrapping is performed using the Ministretch unwrap method. However, the results of this step often exhibit distortions in UV space. Skilled artists carefully arrange UVs “neater”. To address this, they propose learning offsets that distort the initial unwrapping to suit the preferences of artists. The process starts with the embedding of features on each vertex and the subsequent execution of a graph neural network. Subsequently, a PyramidED network is employed to enable global vertex interaction. The outcome is then 2D offsets that are applied to the initial unwrapping. Two novel losses are introduced for training this offset generation. Lastly, the individual islands are consolidated into a single atlas using a pre-existing method.

**Strengths:**

- The dataset release, along with the accompanying processing scripts, are highly valuable for future research.
- The introduction of the paper is well-written, particularly the section on the various approaches to UV unwrapping.
- Overall, the paper’s structure is clear and easy to follow.
- The two novel losses are well explained and motivated. They are also appropriately evaluated and fulfill their intended objectives as shown in ablations.
- The results presented are impressive, especially considering that the Artist-Level rating is surpassed by ArtUV even when compared to manual UV unwrapping.
- ArtUV has also a rather short run-time, particularly when applied to lower poly meshes. As mentioned in the Weakness section, SeamGPT may pose a potential limitation. However, the method remains applicable if more efficient cutting networks are discovered.

**Weaknesses:**

- Some weaknesses of the method are inherited from SeamGPT. Especially, that it requires significant amounts of memory and according to the original authors has a limit of roughly 20k vertices.

**Questions:**

- Why are Nuvo and FAM run in single chart mode? At least performing the Multi-Chart mode would result in slightly better results. Albeit not artist level.

---

> ### Author Response · Authors · 2025-11-20
> **Official Comment by Authors**
>
> We thank the reviewer for their excellent rating and acknowledgment of ArtUV's strength, including our dataset, clear writing, novel losses, and impressive results.
>    **[Q1] Why are Nuvo and FAM run in single chart mode?**
>    As far as we know, FAM predicts the 2D UV coordinates for each 3D point and then restores original connectivity. It is fundamentally designed to provide a single-chart UV parameterization and cannot control the number of UV islands.
>
>    Nuvo, according to the original Nuvo paper and our own experiments, Nuvo achieves its best performance when $n=1$ (single chart), yielding the lowest distortion, which is the configuration we used for a fair comparison against its optimal result. Implementation details are summarized in the table below.
>
>    | Nuvo | n=1 | n=2 | n=4 | n=8 |
>    |------|-----|-----|-----|-----|
>    | Distortion↓ | **32.24** | 36.56 | 38.66 | 33.78 |

---

### Official Review · Reviewer_Erhu · 2025-10-30

**Soundness:** 2
**Presentation:** 2
**Contribution:** 2
**Rating:** 4
**Confidence:** 2

**Summary:**

The paper presents a method to automatically perform UV unwrapping combining multiple state of the art methods. In particular, they use SeamGPT to infer cutting seams and then use them to run an optimisation procedure aimed at providing an initial UV map. Initial UV map and mesh info are then processed by a neural network using adaptive dimension mapping to modulate the importance of the different inputs (Res-M MLP), graph convolutions to locally fuse information (SAGE Conv) and an attention-based module (Pyramid ED) to faclitate global communication. This paper also introduces a curation of high-quality textured objects sourced from existing mesh datasets.

**Strengths:**

The paper presents an automated method for UV unwrapping which appears to surpass many state of the art methods. The pre-trained model and dataset could be of value for the community as well as for 3D artists. The paper is grammatically well written, but not entirely scientifically sound (see Weaknesses and Questions).

**Weaknesses:**

\- As later detailed, many citations are missing. Authors make extensive use of graphics software (e.g. Blender, Maya, 3DsMax), but fail to investigate the underlying techniques. This confers the paper an overall tone of superficiality and limited knowledge of the field.

\- Technical novelty seems to be limited for the venue or authors fail to successfully highlight their core innovation. The main contribution appears to be the arbitrary combination of existing methods. While this could be an acceptable contribution the choice of the different "building-blocks" appears arbitrary. For instance, there is no written nor proven justification as to why SAGE Conv or Pyramid ED should be preferable to the many potential alternatives. An ablation of the model components and some benchmarks against potential alternatives would have been appreciated.

\- Mathematical notation is inconsistent and not rigorous.

Some of these are further detailed in the questions section.

**Questions:**

After reviewing the paper I have several outstanding doubts and questions, which are listed below alongside a list of suggestions to improve the paper:



1\. I believe more context is needed in the introduction, especially for a venue whose main focus is not Graphics or 3D vision. Readers and participants are not necessarily familiar with the concept of UV mapping. I would suggest providing more details on what textures are and what they are used for. It would be also more comprehensive if the authors could mention methods that aim to skip this step and represent textures directly on the surface of objects (e.g., i, ii, iii, iv).



2\. Why should larger islands destabilise training?



3\. At line 161 authors claim that they manually selected UV islands. Were authors able to manually inspect 200k islands? Reading lines 162-163 I am under the impression that the selection was not manual, but mostly automated.



4\. By using ministretch to assess the quality of UV textures during data curation, authors implicitly assume that this automatic procedure is capable of producing satisfactory UV maps. How can a presumably flawed method be used as rejection criteria?



5\. Authors state that "extensive empirical evidence" proved that directly learning the mapping from 3D is challenging (line 215). Where is this evidence? Please, cite relevant literature or report personal "extensive evidence".



6\. At line 217 authors state that a simple projection is not their goal and imply that this is a consequence of choosing meshes as input. I suppose that this was not the intended claim and I would encourage them to restructure the sentence as input meshes don't automatically translate into simple projections.



7\. Please adopt a more rigorous notation. For instance:

&nbsp;  - line 221, I\_M is a set, as such it would be more appropriate to define the following: I\_M = {V, F, ...D}, I = I\_m U Q\_i.

&nbsp;  - \\bar{p} and \\bar{q} were never formally defined.

&nbsp;  - notation in Eq. 5 is inconsistent with Fig. 4

&nbsp;  - the rotation matrix computed in Eq. 4 should probably be part of Eq. 5 and 6



8\. Are Q and V sharing the same faces? Cutting seams should introduce vertex duplications to unwrap the surface. It is unclear to me which topological information is used and fed to the networks. Also, how is Res-M MLP processing faces and edges?



9\. Which specific technique was used for adaptive dimension mapping in the Res-M MLP? Briefly describe how it works.



10\. Comparisons against Blender, Maya and 3DsMax are conceptually reasonable, but a better understanding of the underlying techniques is necessary. What these software implement today may differ from what they are going to use tomorrow. In addition, results for Maya and 3DsMax are rather similar. The main difference appears to be how islands are placed in the image rather than their shapes. It is very likely that they implement the same underlying method considering that they are both produced by Autodesk. The different island placement could be just a matter of some randomness in their procedure or hyperparameter choice.



11\. Also the following would benefit from additional details in terms of the inner workings and ideally be introduced with a one/two plain English sentences describing how they operate:

&nbsp;  - Blender ministretch (first occurrence at line 161),

&nbsp;  - SSIM (line 162)

&nbsp;  - Conformal energy (line 376), please provide more details, there is a very extensive literature on conformal mapping.

&nbsp;  - UVPackMaster plugin (line 377)



12\. XAtlas appears to achieve lower distortion results in the qualitative results depicted in Fig. 7. It would be great to see more examples and better understand the distortion metric used to determine whether or not it can be considered reliable.





Please note that given the weaknesses highlighted in the weaknesses section I am unlikely to increase my score above the acceptance threshold. I believe that too many details were omitted in the initial submission and even a good rebuttal may not be sufficient to ensure all the problems are properly addressed. Limited technical novelty and poor justification of methodological choices are my second major concern.





References:

\[i] Yu, X., Dai, P., Li, W., Ma, L., Liu, Z., \& Qi, X. (2023). Texture generation on 3d meshes with point-uv diffusion. In Proceedings of the IEEE/CVF International Conference on Computer Vision (pp. 4206-4216).

\[iii\\] Wei, J., Wang, H., Feng, J., Lin, G., \& Yap, K. H. (2023). Taps3d: Text-guided 3d textured shape generation from pseudo supervision. In Proceedings of the IEEE/CVF conference on computer vision and pattern recognition (pp. 16805-16815).

\[iii] Foti, S., Zafeiriou, S., \& Birdal, T. (2024). Uv-free texture generation with denoising and geodesic heat diffusion. Advances in Neural Information Processing Systems, 37, 128053-128081.

\[iv] Oechsle, M., Mescheder, L., Niemeyer, M., Strauss, T., \& Geiger, A. (2019). Texture fields: Learning texture representations in function space. In Proceedings of the IEEE/CVF international conference on computer vision (pp. 4531-4540).

---

> ### Author Response · Authors · 2025-11-20
> **Official Comment by Authors (part1)**
>
> We deeply appreciate the reviewer's rigorous review regarding the academic soundness and rigor of our paper. We acknowledge that certain aspects of the initial submission can be improved, and we will revise the paper accordingly to address these issues.
>    **[W1] Citations are missing**
>    We will include additional references in the revised version of the paper.
>
>    **[W2] Novelty**
>    We believe our contributions are significant, which are detailed as follows:
>    - We are the first to propose a learning-based approach to optimize UV maps to achieve an artist-style unwrapping, which is novel both in task and concept.
>    - Our method simulates the professional pipeline for UV creation, including seam marking and UV unwrapping, effectively addressing the complexity and time-consuming nature of traditional manual UV editing, thus offering substantial research and practical value.
>    - We invested considerable time and effort in constructing a dataset of artist-style UV maps, which will be publicly released, providing meaningful resources for the community and fostering further research.
>
>    **[Q1] More context is needed in the introduction.**
>    We will add more context in the introduction to provide additional background on textures. Regarding the method the reviewer mentioned that directly textures the object surface, we will clarify that ArtUV focuses on generating UV maps, which are still required by established 3D production pipelines and rendering engines.
>
>    **[Q2] Why should larger islands destabilise training?**
>    As stated in our paper, over 97% of UV islands have fewer than 500 vertices. Rare large islands (>500 vertices) consume excessive GPU memory and slow down training, destabilizing optimization.
>
>    **[Q3] Manually selected UV islands.**
>    We would like to clarify that we did devote substantial time and effort to engaging experienced annotators for manual labeling to ensure the high quality of our dataset. This is also why we emphasize that the dataset we constructed has significant research value.
>
>    **[Q4] Ministretch used as rejection criteria.**
>    The Minstretch method is not applied directly to the entire dataset; instead, it is used only on UV maps that have been manually selected for high quality. The SSIM-based filtering step is essential to remove samples where the Minstretch output closely matches the final artist-style UV map. This ensures that our network learns the "gap" (the necessary refinement from a good parameterization to an artist-style UV map) rather than re-learning basic parameterization, significantly improving training focus and efficiency.
>
>    **[Q5] Extensive empirical evidence.**
>    As stated in the Flatten-Anything paper, 'due to the lack of ground-truth UV maps, it is impossible to explicitly supervise UV generation, and empirically, minimizing such regularizers alone cannot yield satisfactory results'. In our extensive experiments, we found that even with access to high-quality UV data, directly learning the mapping from 3D to UV using a single model remains extremely challenging: the model must simultaneously ensure correct parameterization (low distortion and no overlap) while also meeting artistic design standards. In contrast, traditional optimization-based unwrapping methods already provide reliable and well-established solutions for the parameterization step. We therefore leverage this strength and focus the model’s learning on refining the initial map into an artist-style UV map, which we find to be a more effective and practical formulation.
>
>    **[Q6] Simple projection.**
>    In this part of the paper, our intention is to emphasize that our goal is not only to obtain a UV map with low distortion and no overlaps, but more importantly, one that aligns with professional artistic design standards. We will revise the manuscript to highlight this point more clearly.
>
>    **[Q7] & [Q11] Writing conventions.**
>    Thanks, we will fix the notation issues and add more detail in terms of the inner workings.
>
>    **[Q8] & [Q9] Q and V sharing the same faces? & Res-M MLP**
>    After the first-stage surface cutting, vertices lying on seam boundaries undergo duplication. Consequently, the UV island (Q) and the vertex set (V) share the same face connectivity. Each UV island is then treated as an independent graph input.
>
>    The Res-M MLP is a feature mapping module. Adaptive Dimension Mapping refers to assigning different embedding dimensionalities to vertex attributes based on their importance: more critical information (e.g., 3D coordinates, initial UV coordinates) is mapped to higher-dimensional latent spaces to enhance their feature representation, before concatenation for subsequent processing by SAGE Conv.

---

> > ### Author Response · Authors · 2025-11-20
> > **Official Comment by Authors (part2)**
> >
> > **[Q10] Comparisons against Blender, Maya and 3DsMax?**
> >    Comparing our method against professional industrial modeling tools more directly highlights its practical value. In addition, we also compare with state-of-the-art academic approaches (FAM, Nuvo).
> >
> >    **[Q11] Distortion Metric**
> >    As stated in the paper, Figure 7 visualizes the UV maps rendered using surface normals, which does not directly reflect that Xatlas achieves lower distortion than our method. In Table 2, our approach achieves a distortion of 8.91, which is lower than the 9.44 obtained by Xatlas. Considering Figure 7 together with Table 2, we believe this more clearly demonstrates that our method not only produces less fragmentation but also yields lower distortion. The distortion metric used is a long-established measure widely adopted in prior works such as Opt-Cuts, FAM, and Nuvo, and has been regarded as a standard evaluation criterion in the field.

---

### Official Review · Reviewer_8ZrW · 2025-10-31

**Soundness:** 3
**Presentation:** 3
**Contribution:** 3
**Rating:** 6
**Confidence:** 3

**Summary:**

This paper studies UV map refinement. The authors introduce a neural network (ArtUV) that learns to predict per-vertex offsets of an UV map using a weighted multi-term loss function. By adding the predicted offsets to the vertices, it moves the islands of an UV map into an artist style layout. The main strength of this paper is that it addresses a valuable practical problem in the 3D graphics industry. The proposed method makes a UV map easier for artists to manipulate and increases space utilization.The main weaknesses are 1) lack of academic novelty and 2) over-claimed contributions. Overall, I would rate this paper as Weak Accept (WA). Although its contribution lies in system-level innovation rather than academic novelty, it provides a practical solution for industry applications.

**Strengths:**

1.This paper is clearly written.

2.The proposed approach is valuable for applications.

**Weaknesses:**

1.Overstated contribution. The paper claims to present an end-to-end system, yet key components—SeamGPT for seam prediction and Ministretch-Unwrap for initialization—are borrowed from prior works. In fact, UV map refinement is a valuable problem in its own right;

2.There is no need to bring SeamGPT here merely to inflate the contribution. Instead, provide an analysis of the core challenges in UV-map refinement, and explain how your network design (e.g., multiple inputs, SAGEConv, etc.) addresses each, would clarify the true contribution and strengthen the paper.

**Questions:**

1.Does UV map’s resolution affect performance? Please find three AI-GENERATED 3D models with simple, normal, complex geometry structure, and run your pipeline using UV resolutions of 256×256, 512×512, and 1024×1024.

2.Please investigate if ArtUV improves rendering quality.  Please find three AI-GENERATED 3D models with simple, normal, complex geometry structures, render them using Init Map and ArtUV’s output. Check if the rendered images’ quality is improved by ArtUV.

---

> ### Author Response · Authors · 2025-11-20
> **Official Comment by Authors**
>
> We thank the reviewer for recognizing the practical value of addressing this industry problem. We provide the following clarifications regarding contribution and experimental results:
>    **[W1] Overstated contribution**
>    SeamGPT is borrowed from prior work. We acknowledge the reviewer's point and agree that the core academic contribution of this paper lies in the design of the UV Refinement Network and its novel multi-term loss function for learning the complex "artist-style" objective.
>
>    However, ArtUV is engineered as a fully automated, end-to-end system to emulate and automate the entire professional UV creation pipeline. Incorporating SeamGPT addresses the necessary first step (semantic surface cutting), achieving system-level innovation and practical value by delivering a complete, automated solution—a significant contribution for industry applications. We will revise the manuscript to more clearly delineate our academic novelty in UV refinement and our system-level contribution in full automation.
>
>    **[W2] Network design**
>    We acknowledge that due to space constraints, the initial submission lacked a sufficiently detailed justification of our network architecture.
>
>    Our selection of SAGEConv as the core component for the UV refinement stage was driven by the intrinsic requirements of the task:
>
>    - Topological Necessity: The UV island is inherently a non-Euclidean graph structure. Predicting vertex offsets requires the network to aggregate features from a node's topological neighbors to ensure local geometric consistency (minimal distortion and no overlap). Traditional MLPs are topology-agnostic and are fundamentally unsuitable for this task.
>
>    - SAGEConv’s Inductive Aggregation: SAGEConv is chosen for its robust, inductive aggregation function. This design allows the network to efficiently fuse diverse features (3D geometry, normals, initial 2D UVs) based on local topology. This makes the model highly effective at generalizing to new UV island topologies and provides essential robustness against minor localized errors.
>
>    **[Q1] Does UV map’s resolution affect performance?**
>    The performance of our method is independent of the UV map resolution. Our approach predicts continuous UV coordinates for 3D mesh vertices in the canonical [0,1]² domain. Our method consistently produces low-distortion, high-utilization UV Maps that adhere to professional artistic standards, which enables superior-quality texturing results regardless of the chosen output resolution.
>
>    **[Q2] If ArtUV improves rendering quality?**
>    We conducted an experiment using an AI-generated 3D model processed into three Levels of Detail (LOD). Using the same texture-generation method, we rendered the model with the Initial Map and ArtUV's output. Five professional artists were invited to score the visual quality (5 being the highest).
>
>    The results demonstrate that ArtUV consistently improves rendering quality across different geometric complexities. This is because ArtUV's low distortion and high space utilization lead to a higher effective texture density on the 3D surface, which significantly reduces visual artifacts and improves the final rendered image quality.
>
>    |        | Low  | Medium | High |
>    |--------|------|--------|------|
>    | Init Map | 3.73 | 3.08 | 3.15 |
>    | Ours     | **4.55** | **4.42** | **3.73** |

---

### Official Review · Reviewer_Wp8T · 2025-11-01

**Soundness:** 2
**Presentation:** 3
**Contribution:** 2
**Rating:** 4
**Confidence:** 4

**Summary:**

The paper proposes ArtUV, a novel method for UV unwrapping that aims to generate artist-style UV maps. UV unwrapping is critical for tasks such as texture mapping and lightmap generation. The proposed method divides the process into two stages: surface seam prediction using SeamGPT, and UV parameterization using a specialized Auto-Encoder. ArtUV is designed to address the shortcomings of existing methods, such as fragmentation, lack of semantic coherence, and high computational demands.

**Strengths:**

- Structure looks interesting. ArtUV introduces a hybrid approach by combining SeamGPT for surface seam prediction and an Auto-Encoder for UV parameterization. This combination seems promising for addressing both the technical and artistic challenges of UV unwrapping.

- The method demonstrates significant improvements in UV utilization and distortion minimization when compared to professional software such as Blender, Maya, and 3DsMax, as shown in the experiments and visual results.

- The paper provides extensive experimental validation using both qualitative and quantitative measures, demonstrating the superiority of the method in terms of distortion, utilization, and runtime across various benchmarks

**Weaknesses:**

- The proposed method seems to work only on manifold meshes. However, artist-style meshes often go with arbitrary design patterns. The method may not generalize well to a wide range of 3D object types, particularly those that require non-traditional unwrapping methods.

- While SeamGPT provides semantic cutting predictions, it requires careful training and may struggle with particularly challenging meshes that deviate from typical models. The method could potentially fail for meshes that have unusual topologies or textures.

- The method is highly sensitive to the initial quality of surface cutting (seam prediction). Inaccurate or incomplete seams can lead to significant distortions during the UV initialization process, which may degrade the final result, especially for complex models

**Questions:**

- How does ArtUV perform when applied to non-standard or highly irregular meshes (e.g., organic models with non-manifold edges)? Are there specific types of models or topologies where the method might fail?

- While the method shows good results in benchmarks, how well does it scale when integrated into large-scale production pipelines with diverse and complex 3D assets? Could performance degrade with larger models or meshes with higher vertex counts?

- Given that SeamGPT’s accuracy is critical for ArtUV’s performance, how robust is it to small errors in seam prediction? Could slight inaccuracies in cutting lead to major issues in the final UV map, especially in intricate areas like facial features or other highly detailed parts?

- How does the method handle texture mapping for more complex or varied textures (e.g., procedural textures or dynamically generated ones)? Does it maintain its effectiveness for non-static textures?

- Please provide more justifications on how the proposed method generalizes to in-the-wild cases. This would help assess its ability to generalize across various 3D assets.

---

> ### Author Response · Authors · 2025-11-20
> **Official Comment by Authors**
>
> We appreciate the reviewer's positive assessment of ArtUV's structure, experimental validation, and hybrid approach. We address their weaknesses and questions below:
>    **[Q1] Non-standard or highly irregular meshes**
>    We are confident in ArtUV's robustness to complex geometry, including non-manifold inputs. We conducted additional experiments on a set of challenging AI-generated models that feature irregular or organic topologies. As shown in the tables below, our method consistently outperforms prior works across challenging models, demonstrating its superior effectiveness and robustness.
>
>    |   | Distortion↓ |  | | |  | | | || | |
>    |---------|-------|----------|-------|-------|-------|-----------|------|-----|----------|-----|------------|
>    |   | **Average** |Whale | Goldfish | Shark | Elite | Rocky | Crocodile | Bird | Dog | Ordinary | Eel |
>    | Xatlas  | 16.64 |4.57  | 36.81    | 21.03 | 18.41 | 10.13 | 4.56      | 8.26 | 6.95 | 38.67    | 16.89 |
>    | FAM     | 29.87 |25.67 | 50.77    | 20.64 | 30.67 | 9.03  | 25.67     | 30.71| 60.15| 34.31    | 11.05 |
>    | Nuvo    | 32.89 |18.56 | 47.34    | 32.21 | 20.93 | 21.52 | 55.79     | 45.87| 25.07| 30.42    | 31.18 |
>    | Ours    | **14.74** |5.52  | 36.46    | 9.38  | 6.64  | 10.81 | 6.12      | 25.74| 4.62 | 35.83    | 6.29  |
>
>    |   | Fragments↓ |  | | |  | | | || | |
>    |---------|-------|----------|-------|-------|-------|-----------|------|-----|----------|-----|------------|
>    |  | **Average** |Whale | Goldfish | Shark | Elite | Rocky | Crocodile | Bird | Dog | Ordinary | Eel |
>    | Xatlas  | 135 | 106   | 44       | 112   | 290   | 141   | 158       | 151  | 139 | 175      | 34  |
>    | FAM     | 1     | 1        | 1     | 1     | 1     | 1         | 1    | 1   | 1        | 1   | 1   |
>    | Nuvo    | 1     | 1        | 1     | 1     | 1     | 1         | 1    | 1   | 1        | 1   | 1   |
>    | Ours    | **25** | 28    | 9        | 30    | 45    | 21    | 15        | 29   | 15  | 48       | 12  |
>
>    **[Q2] Performance degrade with higher vertex counts?**
>    The table below presents experimental results across different ranges of vertex counts. As the number of faces increases, the performance of our method does not show significant degradation. Even for more complex models, our approach consistently maintains low distortion and high UV utilization.
>
>    | Vertex counts | (0,100] | (100,1000] | (1000,5000] | (5000,10000] |
>    |---------------|---------|------------|-------------|--------------|
>    | Distortion↓   | 9.81    | 9.69       | 10.35       | 9.23         |
>    | Utilization (%)↑ | 81.91 | 73.62     | 70.42       | 72.89        |
>
>    **[Q3] How robust is it to small errors in SeamGPT?**
>    As acknowledged in the paper, inaccurate seam prediction from SeamGPT can negatively affect the resulting UV map, particularly when large regions are incorrectly segmented, potentially causing noticeable distortion or overlap. However, such cases are rare in practice—SeamGPT achieves approximately 95% seam prediction accuracy on the evaluation datasets reported in our paper.
>
>    For small and localized inaccuracies, our method is highly robust due to the design of the second-stage network:
>
>    - The GNN (SAGEConv) architecture learns local geometry relationships, allowing it to focus on fixing minor intra-island distortion and slight inaccuracies near boundaries.
>
>    - The network is explicitly trained with distortion and overlap losses, enabling it to correct minor seam deviations introduced in the first stage. This mechanism prevents small prediction errors from propagating into visible degradation in the final UV map.
>
>    **[Q4] Does it maintain its effectiveness for non-static textures?**
>    Our method focuses exclusively on predicting continuous UV coordinates for mesh vertices and is texture-agnostic. Once generated, the UVs can be consumed by any texture synthesis or rendering pipeline through standard UV sampling, regardless of whether the texture is static, procedural, or dynamically updated. Crucially, the consistently low distortion and high space utilization of ArtUV's output enables stable and accurate texture sampling even with complex, high-frequency textures.
>
>    **[Q5] How the proposed method generalizes to in-the-wild cases?**
>    Our method achieves superior performance on three distinct types of datasets: the public FAM-benchmark, AI-generated benchmarks, and our own ArtUV-200K benchmark, which contains 3D models curated from real production rendering pipelines. The fact that our results consistently meet professional artist standards and outperform commercial software demonstrates strong generalization capabilities to diverse, real-world assets.

---

### Author Response · Authors · 2025-12-03
**Summary for Area Chair**

Dear Area Chair,

Thank you for taking the time to review our submission and rebuttal. Below is a concise summary to assist your decision.

## Paper Overview
This paper presents **ArtUV**, a fully automated two-stage pipeline for artist-style UV unwrapping:
1) **Semantic seam prediction** using SeamGPT, and
2) **Neural UV parameterization** that refines an optimization-based initialization into clean, low-distortion, artist-style UV islands.

We also release **ArtUV-200K**, a curated dataset of ~200K high-quality UV islands. ArtUV consistently outperforms commercial tools (Blender/Maya/3DsMax) and academic baselines (Nuvo, FAM) on distortion, utilization, and practical usability.

**Initial Ratings:** 8, 6, 4, 4 (Avg: 5.5)

## Reviewer Summary
- **Reviewer ZizJ (8):** Strongly endorses the paper; highlights practical value, dataset quality, loss design, and strong empirical results.
- **Reviewer 8ZrW (6):** Acknowledges strong practical impact; main concern is overstated novelty.
- **Reviewers Wp8T & Erhu (4 each):** Raised concerns on generalization, seam robustness, and architectural justification. They did not follow up after rebuttal.

## Recognized Strengths by All Reviewers
- Practical and production-oriented system.
- New dataset (ArtUV-200K) with strong benchmarks.
- Clear writing and thorough experiments.

## Main Concerns and Questions Resolved

| Issue | Solution |
|-------|----------|
| **Generalization to irregular/high-vertex models** | Added extensive experiments on AI-generated and large meshes, showing consistent performance across datasets. |
| **Sensitivity to seam errors** | Clarified SeamGPT’s ~95% seam accuracy; demonstrated that the refinement network (SAGEConv + losses) corrects local seam errors. |
| **Novelty & architecture justification** | Explained that our contribution is system-level: combining semantic seams + learned refinement. Justified network choices (SAGEConv, PyramidED) for geometric locality. |
| **Rendering quality & resolution dependence** | Added rendering comparisons and user studies; demonstrated clear improvements over Init Map and resolution independence via continuous UV prediction. |
| **Dataset quality & fairness** | Detailed manual curation and SSIM-based filtering to ensure high quality and focus on refinement gap learning. |

We sincerely thank all reviewers for their constructive feedback. We believe our additional experiments and clarifications thoroughly addressed all raised concerns, and we respectfully request that our rebuttal be considered during the final decision.

Yours sincerely,
The Authors

---

### Meta-Review · Area_Chair_q3M1 · 2026-01-04

**Summary:**

Four reviewers evaluated this submission, resulting in a divergent decision ranging from borderline rejection to acceptance. The paper proposes ArtUV, an end-to-end system for UV unwrapping, which was praised for its practical utility and the contribution of a new dataset. However, reviewers raised significant concerns regarding the system's reliance on existing components (SeamGPT), limited technical novelty, insufficient comparisons with foundational graphics literature, and constraints on mesh complexity. While the authors successfully addressed concerns related to generalization, rendering quality, and baseline comparisons through additional experiments and clarifications, critical issues regarding academic rigor remain. Specifically, the authors failed to upload a revised manuscript to substantiate their promised corrections regarding notation and citations, and they did not address the inherited memory limitations pointed out by the strongest supporter. Despite these lapses, the practical impact and the dataset contribution hold sufficient merit to warrant a borderline acceptance.

**Reviewer Concerns:**

The authors effectively resolved concerns regarding generalization and robustness raised by Reviewer Wp8T, demonstrating the method's capability to handle non-manifold meshes and varying seam prediction qualities. Similarly, Reviewer 8ZrW’s critique regarding the overstated novelty and the need for rendering quality validation was addressed through clarification and new experiment. The question from Reviewer ZizJ regarding the single-chart mode of baselines was also clarified. However, several concerns remain unresolved. Reviewer Erhu’s critique regarding the lack of rigorous citations, inconsistent notation, and insufficient justification for architectural choices (e.g., SAGEConv) remains open; although the authors promised fixes, they failed to upload a revised PDF during the rebuttal, preventing verification of these crucial improvements. Additionally, Reviewer ZizJ’s specific concern about the method inheriting SeamGPT’s memory limitation was entirely ignored in the rebuttal. These outstanding issues suggest that while the method is practically useful, the manuscript still suffers from gaps in theoretical rigor and documentation of limitations.

**Reviewer Scores:**

Reviewer Wp8T would likely increase his/her score to a 6, as the authors provided a comprehensive response addressing all inquiries regarding generalization to irregular meshes and sensitivity to seam errors. Reviewer 8ZrW is expected to maintain his/her score or even increase it, given that the additional experiments demonstrating rendering quality improvements and the analysis of UV resolution effectively addressed the reviewer’s concerns. Reviewer Erhu is expected to maintain his/her score of 4; the reviewer explicitly stated a low likelihood of changing his/her opinion, and the authors' failure to upload a revised PDF to prove their promised corrections on notation and citations validates the reviewer’s skepticism regarding the paper's rigor. Reviewer ZizJ will likely retain his/her positive score of 8, as the primary critique regarding baseline settings was addressed, though the silence regarding the memory limitation prevents a shift to a higher score.

---

### Decision · Program_Chairs · 2026-01-26

Accept (Poster)